# The Multi-Step Chain Extension for Waterborne Polyurethane Binder of *Para*-Aramid Fabrics

**DOI:** 10.3390/molecules27217588

**Published:** 2022-11-05

**Authors:** Ge Ma, Qianshu Wang, Jun Ye, Lifan He, Longhai Guo, Xiaoyu Li, Teng Qiu, Xinlin Tuo

**Affiliations:** 1Key Laboratory of Carbon Fiber and Functional Polymers, Ministry of Education, Beijing University of Chemical Technology, Beijing 100029, China; 2Beijing Engineering Research Center of Synthesis and Application of Waterborne Polymer, Beijing University of Chemical Technology, Beijing 100029, China; 3Key Laboratory of Advanced Materials (MOE), Department of Chemical Engineering, Tsinghua University, Beijing 100084, China

**Keywords:** waterborne polyurethane, chain extension, binder, *para*-aramid

## Abstract

The comprehensive balance of the mechanical, interfacial, and environmental requirements of waterborne polyurethane (WPU) has proved challenging, but crucial in the specific application as the binder for high-performance polymer fiber composites. In this work, a multi-step chain extension (MCE) method was demonstrated using three kinds of small extenders and one kind of macro-chain extender (CE) for different chain extension steps. One dihydroxyl blocked small molecular urea (1,3-dimethylolurea, DMU) was applied as one of the CEs and, through the hybrid macrodiol/diamine systems of polyether, polyester, and polysiloxane, the WPU was developed by the step-by-step optimization on each chain extending reaction via the characterization on the H-bonding association, microphase separation, and mechanical properties. The best performance was achieved when the ratio of polyether/polyester was controlled at 6:4, while 2% of DMU and 1% of polysiloxane diamine was incorporated in the third and fourth chain extension steps, respectively. Under the condition, the WPU exhibited not only excellent tensile strength of 30 MPa, elongation of break of about 1300%, and hydrophobicity indicated by the water contact angle of 98°, but also effective interfacial adhesion to *para*-aramid fabrics. The peeling strength of the joint based on the polysiloxane incorporated WPU after four steps of chain extension was 430% higher than that prepared through only two steps of chain extension. Moreover, about 44% of the peeling strength was sustained after the joint had been boiling for 40 min in water, suggesting the potential application for high-performance fabric composites.

## 1. Introduction

Waterborne polyurethane (WPU) combines the excellent performance of polyurethane with customizable soft–hard segmental structures and the environmentally friendly characteristics of aqueous polymer dispersion [1,2]. WPU is welcomed in various fields like coatings [3] and adhesives [4] as it tends to be safe, easy-to-manipulate, and with low-viscosity. Nowadays, the applications are not limited to everyday items, but have extended to high-performance fabric composites as ideal resin binders [5,6]. The molecular design studies on WPU are thus highlighted, aiming to meet the complex requirements of strength [7], toughness [8], and hydrophobicity [9,10], as well as environmental friendliness [11]. Additional difficulty further come from the dense and hard-to-bind surfaces of the high-performance fibers, which are represented by *para*-aramid fibers of poly(p-phenylene terephthalamide) (PPTA) [12,13]. The exploration on the profound segmental control and microstructure evolution beneath the application performance is then crucial for the development of WPU synthesis [14].

The common polymerization components for WPU are similar to those for polyurethane (PU), including diisocyanate, macrodiol, and small molecular chain extenders (CEs). Among them, CEs are necessary for the extension of the macromolecular chain above the critical entanglement length of polymer. Besides, diverse functions such as hydrophobicity [15], flame retardancy [16], antimicrobial [17], and light emission [18] can also be incorporated into PU via the designs on CEs. In WPU synthesis, hydrophilic CEs like dimethylol propionic acid (DMPA) or dimethylol butyric acid (DMBA) are essential for successful dispersion in water [19]. The formation of branched or chemical/physical crosslinking points is also related to the types of CEs used in WPU synthesis. Although both diol and diamine are selectable for the CEs, small molecules of diols are most preferred. The urea linkages generated by the reaction between diamine and -NCO are sometimes preferred as they would provide more ordered H-bonding sites for physical crosslinking than the carbamate linkages generated by the reaction of the diol extenders [20,21,22,23,24,25]. However, too fast reaction rates of small molecular diamine would also increase the difficulty in the control of the reaction, as well as the subsequent dispersion stability.

Besides the CEs, another design point is on the macrodiol. The soft segments introduced by the copolymerization of the macrodiol with -NCO take the highest mass content in the WPU polymer. Different types of macrodiol such as polyether, polyester, or polycarbonate contribute to pronounced property diversity. Alternatively, the mixture of the different kinds of macrodiol can also be applied in one WPU synthesis [26,27]. In the work of Lee et al., two kinds of polyether diol were mixed under different ratios for the preparation of a WPU-based textile coating with excellent water repellency [28]. It has been revealed by Cakić et al. that the phase separation, crystallinity, and thermal stability were all affected by the ratio, length, and oxygen content of the macrodiol of poly(ethylene glycol) (PEG) or poly(propylene glycol) (PPG) in the mixture with the polyester macrodiol of polycaprolactone (PCL) [29] or polycarbonate diol (PCD) [30], providing a clue to fine-tune the performance of WPU adhesives. The mixture effect of PCD and polyester macrodiol has been reported by Martín-Martínez et al. [31]. They found that the mean particle sizes of the WPU dispersion increased while the viscosity decreased when the macrodiol was applied in the mixture form. The complex interactions between the different types of segments were reflected by the degree of phase separation and the adhesive strength for chlorinated vulcanized styrene-butadiene rubber/WPU/roughened leather joints. However, the influence of the macrodiol mixture on the WPU system with a urea block strengthening H-bonding network is still not reported in recent research. Moreover, aiming to further improve the water resistance and flexibility, the incorporation of siloxane moieties by chain extension is also an effective method [32]. However, the dramatic sacrifice in the adhesion strength would be the drawback when the product is to be applied as a binder for fabrics.

It should be noted that the desirable comprehensive performance is not going to be achieved by only a single type of CE or macrodiol, but will depend on the holistic optimization on the multi-segmental structures, which can be integrated via the stepped chain extension process so that the different characteristics brought by different component would be mostly compatible. Following this clue, we demonstrate here a multi-step chain extension (MCE) method for WPU preparation. In the first and second steps of chain extension, the popular small molecular CE of DMBA and 1,4-butanediol (BDO) was used separately. Another small molecular diol of 1,3-dimethylolurea (DMU) was introduced as the CE for the third chain extension step. The specialty of DMU is that the urea group is inside the molecule with the active hydroxyl groups at the both ends serving as the reactive sites. The chain extension of diol with -NCO blocked oligomer not only has a proper reaction rate to ensure the steady chain propagation and the following dispersion, but also makes it easy to act in concert with other hydroxyl-based chain extenders for the MCE method. In this way, the steady MCE method showed high tolerance to the component variation. Thanks to this advantage, the macrodiol mixture of polytetramethylene ether (PTMG) and polycaprolactone glycol (PCL) with different ether/ester ratios was adopted in the synthesis. As a further expanding of the robust MCE, etherimide-blocked polydimethylsiloxane (PDMS) oligomers were incorporated as the final CE for the fourth step of chain extension. The synergism of the urea groups introduced by DMU with different segments are revealed as responsible for the distribution control on the H-bonding-associated hard domains in the soft matrix, as well as the strength, roughness, and interfacial adhesion to *para*-aramid fabrics of the WPU. The stable PDMS-modified WPU obtained showed improved hydrophobicity without the loss of a high level of strength, toughness, and adhesion strength. To illustrate the different contribution of the CEs and soft segments in MCE, the following discussion is mainly divided into four parts. Section 2.1 and Section 2.2 clarify the influence of DMU on WPU when single types of macrodiol were used. The influence of the types of CEs on the different steps of MCE is discussed in Section 2.3 and Section 2.4. Section 2.5 discusses the influence of the macrodiol mixture on the MCE. Finally, the adhesive properties of WPU are studied in Section 2.6.

## 2. Results and Discussion

### 2.1. Chain Extension by DMU

The effectiveness of DMU as the CE to react with -NCO in isophorone diisocyanate (IPDI) was verified by the model reaction of IPDI and DMU under the molar ratio of 1:2 catalyzed by DBTDL at 50 °C. The reaction was monitored by FTIR, with the results shown in Figure 1. The original peak height of -NCO at 2275 cm^−1^ is set as *x*, the peak height of -NCO consumed in each period is set as *x_n_*, and the conversion rate *p* in each period is *x_n_/x*. In Figure 1a, the intensity of the band at 2275 cm^−1^ characteristic for -NCO gradually decreased and finally disappeared during the 120 min of the reaction. The kinetics can be revealed by the time-related evolution of the conversion degree of the functional groups in the polymerization process (*P*) shown in Figure 1b, where the profile is well fitted to the second-order kinetics. The reaction rate constant *k* and the correlation coefficient *R*^2^ were derived by linear fitting, shown as the inset of Figure 1b, which was 6.0 × 10^−2^ mol^−1^·L·min^−1^ and 0.95, respectively.

As the reactivity was verified, DMU (D) was then applied in MCE, sketched as Figure 1. The formulations of the different reactions as well as the naming of different samples are summarized in Table 1.

### 2.2. Impact of DMU Content on WPU

The influence of the DMU content (*p*) on 3D*_p_*-G is illustrated in Figure 2, where DMBA, 1,4-butanediol (BDO), and DMU (D) were used as the CEs for the first, second, and third chain extension steps and denoted as CE-1, CE-2, and CE-3, respectively. PTMG (G) was used as the macrodiol is this part. IPDI was applied as the diisocyanate for all of the reactions involved in this work. The dispersion state of the prepared WPU is shown in Figure 2a. It can be seen that, with the increase in *p*, the dispersion changed slightly from light transparent to milky white. After storage at room temperature for up to 6 months, the apparent appearance of all dispersions remained basically unchanged, indicating good stability. The DLS results of the samples of 3D*_p_*-G are provided in Appendix A. The average diameters were in the range of 56~86 nm, with the PDI between 0.12 and 0.35. The ATR-FTIR spectra of the WPU samples are shown in Figure 2b. Typical peaks of almost all of the IPDI-PTMG types of PU can be observed in the spectra, including the bands at 2954 cm^−1^ and 2877 cm^−1^ for the C-H vibration of CH_2_ and CH_3_, 1600~1795 cm^−1^ for C=O stretching vibration, 1510 cm^−1^ for N-H bending vibration, 1436~1480 cm^−1^ for the bending vibration of CH_2_ and CH_3_, 1263 cm^−1^ for C-O stretching vibration, and 1116 cm^−1^ for C-O-C aliphatic ether.

The influence of the urea groups introduced by DMU can be seen on the changes of the peak shapes of the carbonyl bands, which broaden to the lower wavenumber range with the copolymerization of DMU in Figure 2b. The curve fitting results on the carbonyl bands are displayed in Figure 2c. For the sample of WPU-G, three sub-peaks can be divided at about 1720 cm^−1^, 1700 cm^−1^, and 1655 cm^−1^, which are assigned to the free, disordered, and ordered H-bonded C=O, respectively [31,33,34,35,36]. The contributions of the three kinds of C=O were estimated by the ratios of peak area integration and are listed in Appendix A. By the integration on the area of the different sub-peaks, the H-bonding degree and the contribution of the different types of H-bonds can be calculated through the following equations:Contribution(*n*) = *A_n_*/(*A*_1_
*+ A*_2_
*+ A*_3_) × 100%(1)
where *A_n_* is the area of the subpeak and *n* = 1, 2, or 3 denotes the free, disordered and ordered H-bonded C=O, respectively.

It can be seen clearly that the contribution of free C=O was obviously suppressed by the addition of DMU. The initial small amount of DMU (*p* ≤ 2) would mainly affect the disordered H-bonding. However, when *p* increased to 3 and above, there would be the competition of the three kinds of C=O caused by the allocation of the additional C=O in urea groups. However, the H-bonding degree was maintained at higher levels in comparison with that of WPU-G. This should be the reason for the increased mechanical properties of the 3D*_p_*-G samples.

The H-bonding association is related to the microphase separation of PU [37], which can be characterized by DMA. In the spectra of *E’* versus temperature in Figure 2d, the glass platforms of all of the profiles overlapped seriously in the low temperature region and sharp differences were observed after the glass transition. In comparison with the sample of WPU-G, the modulus of the rubber elastic platform was elevated obviously for the samples of 3D*_p_*-G, suggesting the increase in physical crosslinking via the H-bonding association prompted by DMU. In Figure 2e, all of the 3D*_p_*-G films exhibited two glass transition peaks in the spectra of tan*δ* versus temperature. The peak around −50 °C corresponding to the glass transition temperature of the soft segments (*T*_g,s_) was basically unchanged with the increase in *p*. The phenomena indicate that DMU would act mainly on the hard segmental domains, which is evidenced by the gradual movement of the peak for the glass transition temperature of the hard segments (*T*_g,h_) toward higher temperature ranges. At the same time, the distance between the two peaks (Δ*T* = *T*_g,h_ − *T*_g,s_) increased, while the intensity of the tan*δ* decreased in this saddle region. All of these *p*-related results illustrate clearly that the addition of DMU as CE-3 in the MCE preparation of WPU promoted the microphase separation of the hard segments from the soft matrix.

The H-bonding associated microphase separation would then impact on the mechanical properties [38]. The typical tensile curves are shown in Figure 2f. It is not surprising to see that the tensile strength of the WPU films was significantly improved by the 1% of DMU addition (*p* = 1). As *p* increased from 1 to 5, the slope of the curves went up while the elongation at break decreased steady, which can be understood as the consequence of prompted physical crosslinking. The optimal results were achieved at the point of *p* = 2, which gave the highest strength of 29.0 MPa with the elongation at break of 1045%. Without the third chain extension by DMU, the tensile strength of the film was only 6.13 MPa with an elongation at break of 1224%. Comparatively, 2% of DMU used as CE-3 provided a 473% strength improvement, with the elongation at breaking maintained above 1000%. The variation in the tensile strength and elongation at break as a function of *p* is provided in Appendix A. The change in the toughness of the WPU films can be reflected by the fracture energy. In Figure 2g, there was an elevation in the fracture energy of WPU accompanied with the addition of DMU, and its maximum value of 101 MJ·m^−3^ also appeared when *p* = 2. The results show that the introduction of DMU as CE-3 has the effect of strengthening and toughening the WPU.

### 2.3. Influence of CE-3 Types

The contribution of DMU to the structure and properties of WPU can be further illustrated by the comparison with other small molecular CEs. In Figure 3, three kinds of CE-3 as DMU, TMP (T), and PTL(P) were separately used in the preparation of 3D_2_-G, 3T_2_-G, and 3P_2_-G, respectively. The three kinds of dispersion obtained were all uniform, stable, and with good fluidity. The number-average particle size and PDI can be seen in Appendix A. The average diameters were between 47 and 68 nm with the PDI in the range of 0.18~0.30.

The FTIR spectra are provided in Appendix A. The curve fitting results on the carbonyl region are shown in Figure 3a and the quantitative results are summarized in Appendix A. It can be seen that the carbonyl bands of 3T_2_-G and 3P_2_-G shifted obviously to the higher wavenumber range compared with 3D_2_-G. Different from the sample of 3D_2_-G, where the free and disordered hydrogen bonded C=O played important roles, the films of 3T_2_-G and 3P_2_-G were dominated by ordered hydrogen bonding.

The improved ordered degree suggests the deepening localization of hydrogen bonds, which would sacrifice their contribution to the overall physical crosslinking networks [38]. It can be seen in the *E*’~temperature spectra in Figure 3b that the 3T_2_-G sample with the highest ordered H-bonding degree exhibited the lowest modulus in the whole testing temperature range. The other two samples exhibited similar modulus at the glass platform. However, after the glass transition, the modulus dropped more rapidly for 3P_2_-G than that for 3D_2_-G. The difference suggests again that there would be more physical crosslinking in 3D_2_-G films with relatively good thermal resistance contributed by the highest content of disordered H-bonding structures.

In Figure 3c, two glass transition peaks are shown in the tan*δ*~temperature spectra. The *T_g_*_,s_ of 3D_2_-G, 3T_2_-G, and 3P_2_-G were all around −50 °C. However, the *T_g_*_,h_ shifted to the high temperature regions in the sequence of 3P_2_-G (24 °C), 3T_2_-G (31 °C), and 3D_2_-G (39 °C). Judging by the value of Δ*T*, the 3D_2_-G film would be of the highest degree of microphase separation, which tends to present balanced comprehensive tensile properties. In Figure 3d and Appendix A, the tensile strength of 3D_2_-G was almost 6.7 and 13.2 times that of 3T_2_-G and 3P_2_-G, respectively. The enhanced strength can be related to the reinforcement of the relatively well developed hard domains by the synergism of disordered and ordered H-bonding association. Consequently, the elongation at break of 3D_2_-G is lower than that of the other two groups of samples. According to the literature, a proper amount of disordered H-bonded C=O would help for the stress release and elongation orientation, which is important for the elevation in mechanical strength [39]. Moreover, free C=O is also necessary for interfacial adhesion. In this way, the balance of the three kinds of C=O is the key for the desired optimal properties.

### 2.4. Influence of the Addition Sequence of DMU

The optimization on the H-bonding association not only relies on the types of the CEs, but their rational distribution on the backbones controlled by the reaction sequence of the stepped polymerization. The influence of the addition sequence of DMU in MCE is illustrated in Figure 4. Three samples of 1D_2_-G, 2D_2_-G, and 3D_2_-G were prepared with DMU added as CE-1, CE-2, and CE-3, and the chain extension sequence is DMU-DMBA-BDO, DMBA-DMU-BDO, and DMBA-BDO-DMU, respectively. In all three CE addition manners, stable WPU dispersion was obtained, with their average particle diameters and PDI displayed in Appendix A. The number-average particle diameters were between 68 and 79 nm and the PDI was between 0.23 and 0.32. However, the balanced contribution of both ordered and disordered H-bonds can only be achieved by the sample of 3D_2_-G with DMU serving as CE-3, as indicated in Figure 4a and Appendix A. The FTIR spectra in the wavenumber range from 4000~400 cm^−1^ are shown in Appendix A. The *E*’ versus temperature spectra can be found in Figure 4b. Again, the sample of 3D_2_-G showed the highest *E’* in the whole temperature range. With the postponed addition of DMU from the first to the second and then the third batch of CE, the descending slopes of the curves after glass transition descended in the sequence of 1D_2_-G, 2D_2_-G, and 3D_2_-G. The deceleration would possibly be explained in that, the greater the spacing from the soft segments, the greater the contribution DMU would provide to the microphase separation [40].

Further proof is provided by Figure 4c. In the tan*δ*~temperature spectra, the largest Δ*T* was observed on the sample of 3D_2_-G. The sample of 1D_2_-G showed the lowest *T*_g,h_ and the slightly high-temperature shifted *T*_g,s_, assigning to the poorest developed microphase separation when DMU was located too close to the macrodiol segments. For the sample of 2D_2_-G, both the *T*_g,s_ and *T*_g,h_ moved in the same direction toward high temperatures. We suppose that the H-bonding associated physical crosslinking would play the dominating role when DMU was introduced before BDO but after DMBA.

The effect on the physical crosslinking can also be seen in Figure 4d and Appendix A. Compared with the tensile strength and elongation at break of 1D_2_-G, the tensile strength and elongation at break of 2D_2_-G increased, as seen in general cross-linked systems. However, the optimal strength and toughness will not be achieved until the sufficient microphase separation has been reached beyond the physical crosslinking, which is only in the case described by 3D_2_-G.

### 2.5. Influence of the Macrodiol

PTMG (G) and PCL (L) have different structures and characteristics. Therefore, it is a rational design to use the mixture of them in MCE so that the performance can be further improved and balanced. The influences of the different mixture ratios on the particle size and PDI are provided in Appendix A. The number-average particle sizes were in the range of 48~75 nm and the PDI of the particles were in the range of 0.22~0.36. All of the dispersions were slightly white and translucent with good fluidity. After 6 months of storage at room temperature, the apparent appearance of the dispersion was maintained, indicating good stability. The FTIR spectra of the WPU films are provided in Appendix A. The typical curve fitting results on the carbonyl region are shown in Figure 5a and the quantitative results are summarized in Appendix A. It can be seen that, with the addition of PCL, the bands broadened obviously to the higher wavenumbers contributed by the ester C=O groups, as seen in the sample of 3D_2_-G_6_L_4_. The further increased content of PCL resulted in the access of free C=O indicated by the strengthened band at ~1750 cm^−1^ in the sample of 3D_2_-G_6_L_4_. The strongest band of free C=O was detected in the sample of 3D_2_-L when all of the macrodiol of polyether was replaced by polyester.

In the DMA storage modulus tests in Figure 5b, it can be seen that, in the process of temperature elevation from −100 °C to 100 °C, the modulus after the glass transition dropped faster and faster with the increased PCL content until the maximum was reached at the point of *x*/*y* = 6/4. We ascribe the enhanced temperature sensitivity to the imperfect hybrid microcrystalline domains of polyester [41], whose melting resulted the accelerated dropping of the modulus after the glass transition. For the sample of 3D_2_-G*_x_*L*_y_*, another transition to viscous flow was clearly detected at the temperature close to 100 °C. The identical viscous flow transition stage would benefit the thermal activation of the WPU film in thermoplastic adhesive applications. With the further increase in the content of PCL, increasing development of the micro-domains of polyester crystalline was reflected in the overall elevation on the modulus curves.

More information on the microphase separation is provided in the tan*δ* spectra in Figure 5c. In the figure, the *T*_g,s_ of the polyether segments in 3D_2_-G was observed at −46 °C, while the *T*_g,s_ of the PCL segments in 3D_2_-L was detected at −18 °C, 28 °C higher than that of 3D_2_-G. For the samples of 3D_2_-G*_x_*L*_y_*, the *T*_g,s_ right-moved steady from the −39 °C of 3D_2_-G_8_L_2_ to −23 °C of 3D_2_-G_2_L_8_ as *x*/*y* decreased, just within the temperature range with the upper and lower boundaries determined by the *T*_g,s_ of 3D_2_-L and 3D_2_-G, respectively. In this way, the movement of the *T*_g,s_ peak can be explained simply by the mixture composition of the soft segmental domains.

In the case of *T*_g,h_, both 3D_2_-G and 3D_2_-L showed similar temperatures of 40 °C. The sample of 3D_2_-L tended to behave with higher tan*δ*. The additional energy loss would be from the higher interaction of the ester groups in comparison with the flexible ether groups. The intensities of the tan*δ* peaks for 3D_2_-G*_x_*L*_y_* were also located reasonably between those of 3D_2_-G and 3D_2_-L. However, their peak temperatures moved to a temperature lower than that of either of 3D_2_-G or 3D_2_-L. Moreover, the shape broadening on the peak was also detected. The broadened and left-moved peaks indicate the formation of large contents of some mediated hybrid domains of urethane with some of the mixed macrodiol segments. The contribution of the hybrid reached its maximal at *x*/*y*= 6/4, which can be used to explain the lowest *T*_g,h_ of 23 °C, a relatively strong peak with an intensity comparable to that of 3D_2_-G and a sharpened shape with the narrowest peak width. Since then, the further decrease in *x*/*y* tended to push the broadening of the *T*_g,h_ to high temperatures again.

It can be seen in Figure 5d and Appendix A that, with the increase in the PCL ratio, the tensile strength of the WPU film increased, among which the tensile strength of 3D_2_-L reached 53 MPa. The elongation at break, on the other hand, remained around 1000% until *x*/*y* decreased to 4/6. An exceptional elongation at break higher than 1200% could be explained by the plasticizing effect of free C=O. However, the mobility of this part of C=O would possibly be more and more restricted by the microcrystalline regions formed by PCL with further increased contents. Consequently, the elongation at break decreased again when the tensile strength increased. Moreover, it is worth noting that the largest elongation at break is shown by 3D_2_4M_1_-G_6_L_4_. Through the four-step chain extension reactions by DMBA-BDO-DMU-PDMS, the elongation at break of the WPU film increased close to ~1300% (1295%). Meanwhile, although there is some loss due to the introduction of low-cohesion PDMS segments, the mechanical strength remains at 30.4 MPa, which is comparable to 3D_2_-G. Compared with 3D_2_-G, the toughness was also improved and the fracture energy reaches 122.1 MJ·m^−3^. Besides, the water hydrophobicity is also improved by the siloxane modification [42]. In Figure 5e, the water contact angles (WCAs) of WPU-G were about 74°, while the WCA of the sample of 3D_2_4M_1_-G_6_L_4_ increased to about 98°.

### 2.6. Properties as Fabric Adhesives

To further illustrate the advantage of MCE, which can provide balanced comprehensive properties, we focus here on the application of the product as an aramid fabric adhesive, which is one challenge in the work on adhesions. The good strength, toughness, and flexible PDMS block-related properties also suggest the product to be a good fabric binder. Herein, *para*-aramid fabrics are selected as the substrate, which were adhered together by hot-pressing using WPU prepared in this work as the binder. The adhesive strength of the joint is displayed in Figure 6a. It can be seen that the average peel strength of WPU-G is 0.74 N·cm^−1^, while the average peel strength of 3D_2_-G with DMU chain extension increased to 1.97 N·cm^−1^ and the average peel strength of 3D_2_-G_6_L_4_ copolymerized with PTMG and PCL at 6:4 was 3.91 N·cm^−1^. The improvement in the binding strength when using DMU as CE-3 and the hybrid macrodiol of PTMG and PCL under the ratio of 6:4 was so effective that it would not be much interrupted by the additional introduction of PDMS blocks via the copolymerization of PDMS as CE-4. The average peel strength of 3D_2_4M_1_-G_6_L_4_ was 3.16 N·cm^−1^, which was only slightly lower than that of 3D_2_-G_6_L_4_. The further water resistance was tested under the tedious condition by boiling the joints in water for 40 min. The adhesion remained effective during the boiling process without the observation of the disintegration of bonded structures. The binding strength was tested after the joints were dried at room temperature. It can be seen in Figure 6b that the average peeling strength dropped to 1.72 N·cm^−1^, about 44% of the initial levels. The observation of the fracture section suggested that all of the fractures occurred by the detachment at interfaces.

## 3. Materials and Methods

### 3.1. Materials

Isophorone diisocyanate (IPDI) was obtained from Bayer (Hubei, China)Chemie A.G. Polytetramethylene ether glycol (PTMG) and polycaprolactone glycol (PCL), both of the number-average molecular weight of 2000 g/mol, were provided by Jiangsu Jiaren Chemical Co., Ltd. (Nantong, China) and dried in a vacuum oven at 85 °C for more than 24 h before use. The chemicals of 2,2-dimethylolbutyric acid (DMBA, Shanghai Dibo Biotechnology Co., Ltd., Shanghai, China), 1,4-butanediol (BDO, Tianjin Guangfu Chemical Reagent Factory, Tianjin, China), 1,3-dimethylolurea (DMU, Shanghai Bide Pharmaceutical Technology Co., Ltd., Shanghai, China), trimethylolpropane (TMP, Beijing Chemical Factory, Beijing, China), pentaerythritol (PTL, Tianjin Fuchen Chemical Reagent Factory, Tianjin, China), poly(dimethylsiloxane) etherimide (PDMS, number-average molecular weight of 1000 g/mol, Aladdin Reagent Co., Ltd., Shanghai, China), triethylamine (TEA, Tianjin Fuchen Chemical), and dibutyltin dilaurate (DBTDL, Tianjin Guangfu Fine Chemical Co., Ltd., Tianjin, China) were used as received. *N*-methylpyrrolidone (NMP, Tianjin Fuchen Chemical) was dried with molecular sieves for more than 48 h before use. Deionized (DI) water self-made in the laboratory was used in the preparation.

### 3.2. Preparation of WPU Dispersion

The detailed formulations are summarized in Table 1. The process is sketched in Figure 1. Specifically, in a four-necked flask connected with mechanical stirring, a nitrogen inlet, and an outlet, quantitative PTMG/PCL and IPDI were sequentially added, followed by the dropwise addition of the appropriate amount of catalyst DBTDL. After reacting at 80 °C for 30 min, the first batch of chain extender (CE-1) of DMBA was added as the solution in 5 mL of NMP. After reacting at 80 °C for 2.5 h, the second batch of chain extender (CE-2) of BDO was added. One hour later, the third batch of chain extender (CE-3) of DMU was added for another 1 h of reaction. After reducing the temperature to 50 °C, PDMS was added and the reaction was again prolonged for 1 h. To simply the expression, the PDMS here was denoted as the fourth batch of chain extender (CE-4) despite the reactive difference from the small molecular extenders. TEA with an equal molar ratio of DMBA was added for neutralization. After the neutralization was completed in 30 min, the temperature was lowered to room temperature and deionized water was gradually injected into the reactor under vigorously stirring speed of 2000 r/min for about 30 min to obtain WPU dispersion.

### 3.3. Film Formation

An appropriate amount of WPU dispersion was cast into a silicon mold (8 cm × 6 cm) for film formation at room temperature for 3 days. The films were then dried in a strong convection oven at 70 °C for 12 h. Finally, films with a thickness of about 0.4 mm were stored at room temperature for tests.

### 3.4. Adhesive Joint Preparation

The adhesive properties of WPU were tested according to GB/T 2791-1995 and the substrate was plain-weaved *para*-aramid fiber fabric. The fabric was cut into splines to a size of 200 mm × 25 mm^2^ and the effective bonding area was 150 × 25 mm^2^. The surfaces of the splines were thoroughly cleaned with anhydrous ethanol. After that, the splines were immersed in the WPU prepared in this work for 30 s, hung in the air for 10~20 min, and dried in an oven at 120 °C for 0.5 h. The splines were hot pressed at 60 °C for 30 s until they were smooth. One side of the spline was coated with 2 mL of WPU dispersion and placed in a 120 °C oven for 1 h to dry. With the face-to-face stack of the WPU applied surfaces, the two pieces of splines were pressed together at 120 °C for 120 s. The bonded fabric joints were dried in an oven at 120 °C for 1 h, hot pressed under 10 MPa at 120 °C, and cooled at room temperature for more than 24 h before peeling tests.

### 3.5. Characterization

The particle size (in number average diameter) and particle size distribution (PDI) of the dispersion were measured by dynamic laser scattering (DLS) via a Zetasizer Nano ZS, (Malvern, UK) Fourier transform infrared (FTIR) spectra were collected by a Tensor 37 (Bruker, Germany) spectrometer with a scanning resolution of 4 cm^−1^, number of scans of 16, and scanning wavenumber range of 4000~400 cm^−1^. Attenuated total reflection Fourier transform infrared spectroscopy (ATR-FTIR) measurements were performed on a Nicolet iS5 infrared spectrometer (The United States,Thermo Scientific, Waltham, MA, USA) with an iD7 attenuated total reflection accessory (ZnSe) crystal. Spectra were collected in the range of 4000~550 cm^−1^ at an ambient temperature of 25 °C using 64 scans. The tensile properties of the films were tested according to the method specified in GB/T 1040.3-2006. A universal testing machine of CMT4304 (Shanghai, China) was applied for this purpose with the speed of the machine head controlled at 50 ± 5 mm/min. The T-peel strength of the adhesive joints was also tested on the same machine at a peeling rate of 50 mm/min. The dynamic mechanical analysis (DMA) was carried out on a Q800 (TA) under the film tension mode at the fixed frequency of 1 Hz when the temperature was increased from −100 °C to 150 °C with a ramping rate of 3 °C/min. A weak force of 0.01 N was preloaded on the film sample before the test.

## 4. Conclusions

With the MCE method, using DMBA, BDO, DMU, and PDMS as the CE for the first, second, third, and fourth steps of chain extension, stable WPU were prepared with the particle sizes in the range of 56~86 nm, which varied depending on the specific compositions. The chain extension of the di-hydroxyl groups of DMU was very compatible with the reaction system and provided a reliable method for the building of urea linkage in the backbone of WPU without a sacrifice in the reaction control and dispersing stability. The additional urea linkages contributed to the increased H-bonding association degree, which prompted the microphase separation during the film formation of WPU especially when 2% of DMU was applied as CE-3. At the same time, the mixture of PTMG with proper amount of PCL using the macrodiol in WPU polymerization tended to lead to the specific hybrid phase when the mixture effect was combined with the effect of DMU. Moreover, taking advantage of the robust preparation, a small amount of di-etherimide blocked PDMS was also incorporated in the WPU. Under the optimal condition, the sample of 3D_2_-G_6_L_4_ exhibited a tensile strength of 44 MPa and elongation at break of 983% with a break energy of 131.4 MJ·m^−3^. The polymerization of PDMS contributed to additional elongation at break (~1300%) and hydrophilicity (WCA = 98°), while the strength was maintained at high levels of 30 MPa. The as-synthesized WPU is further used as a fabric binder for *para*-aramid. Although limited by poor interfacial conditions, an improvement in the adhesive strength from 0.74 N·cm^−1^ to 3.91 N·cm^−1^ was also achieved with 44% of maintenance after boiling for 40 min in water. All of the results indicate promising potential applications as environmentally friendly resin binders for high-performance polymer fabric composites.

## Data Availability

The data presented in this study supporting the results are available in the main text. Additional data are available upon reasonable request from the corresponding author.

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
