# Peer review of "The Multi-Step Chain Extension for Waterborne Polyurethane Binder of Para-Aramid Fabrics"

_molecules, 2022, doi:10.3390/molecules27217588_

Round 1

Reviewer 1 Report

The research article entitled “The multi-step chain extension for waterborne polyurethane

binder of para-aramid fabrics” revealed a promising idea to improve the specific properties of polymer fiber composites. This paper demonstrated a very good scientific concept for the journal's scope. So, I recommend a minor revision of this manuscript as follows.

- Figure 1a, please reduce the x scale to improve the visibility of the spectrum. (2000-2500) because the authors didn’t explain the other zone.

- Figure 1b, please explain how to determine the conversion degree of -NCO in polymerization (P)

- Figure 2b, please reduce the x scale to improve the visibility of the spectrum. (1500-2000) because the authors didn’t explain the other zone.

- please explain how to determine the Impact of DMU content on WPU

- Do you have any idea for the change of chain extension on other properties such as thermal properties (by TGA/DSC), please discuss with literature review.  

Author Response

请参阅附件。

Round 2

Reviewer 2 Report

the revised paper is OK